# Designing narcissistic self-sorting terpyridine moieties with high coordination selectivity for complex metallo-supramolecules

Jianjun Ma [1], Tong Lu[1], Xiaozheng Duan[2], Yaping Xu [1], Zhikai Li [3], Kehuan Li [1], Junjuan Shi [1], Qixia Bai [4], Zhe Zhang[4], Xin-Qi Hao [5], Zhi Chen[3], Pingshan Wang [4] & Ming Wang [1]✉

Coordination-driven self-assembly is a powerful approach for the construction of metallo-supramolecules, but designing coordination moieties that can drive the self-assembly with high selectivity and specificity remains a challenge. Here we report two ortho-modified terpyridine ligands that form head-to-tail coordination complexes with Zn(II). Both complexes show narcissistic self-sorting behaviour. In addition, starting from these ligands, we obtain two sterically congested multitopic ligands and use them to construct more complex metallo-supramolecules hexagons. Because of the non-coaxial structural restrictions in the rotation of terpyridine moieties, these hexagonal macrocycles can hierarchically self-assemble into giant cyclic nanostructures via edge-to-edge stacking, rather than face-to-face stacking. Our design of dissymmetrical coordination moieties from congested coordination pairs show remarkable self-assembly selectivity and specificity.

[1] State Key Laboratory of Supramolecular Structure and Materials, College of Chemistry, Jilin University, Changchun, Jilin 130012, China. [2] State Key Laboratory of Polymer Physics and Chemistry, Changchun Institute of Applied Chemistry, Chinese Academy of Sciences, Changchun, Jilin 130022, China. [3] College of Chemistry and Environmental Engineering, Shenzhen University, Shenzhen, Guangdong 518060, China. [4] Institute of Environmental Research at Greater Bay Area; Key Laboratory for Water Quality and Conservation of the Pearl River Delta, Ministry of Education; Guangzhou Key Laboratory for Clean Energy and Materials; Guangzhou University, Guangzhou, Guangdong 510006, China. [5] Green Catalysis Center, Henan Key Laboratory of Chemical Biology and Organic Chemistry, and College of Chemistry, Zhengzhou University, Zhengzhou, Henan 450001, China. ✉email: mingwang358@jlu.edu.cn

Coordination-driven self-assembly has been proven to be an effective approach for constructing supramolecular structures, due to its high predictability[1–6]. As such, a variety of discrete metallo-supramolecular architectures have been fabricated through different types of self-assembly strategies, including the coordinations of pyridyl[7], bipyridyl[8], terpyridyl[9], or heterotopic ligands[10–12] with naked metal ions or metal-organic components[13,14]. Given the increasing demands for diverse and predesigned molecular structures and functions, scientists are encouraged to further develop novel building blocks to achieve high coordination selectivity in precise and controllable supermolecular assembly.

2,2′:6′,2″-tpy and its derivatives have been widely employed in coordination-driven self-assembly on account of their variable coordination ability with transition metal ions, as well as the unique properties and wide applications in optical devices[15], catalysis[16], self-healing[17], and drug delivery[18] after complexation. Among the design of tpy-based metallo-supramolecules, the most common strategy is connecting tpy units and directing units at the central pyridine for subsequent coordination, which usually caused the poor selectivity of coordination units[4,19–23]. To further enhance geometrical diversity and complexity of assemblies, a hierarchical stepwise assembly strategy has been developed to promote the controllability of the self-assembly process by using metals that can form strong coordinative bonds with tpy, such as <tpy-Ru(II)-tpy> or <tpy-Fe(II)-tpy>[9,24,25]. However, the tedious synthesis and column separation procedures limit its extensive application in supermolecular constructions[25–27]. Currently, it still remains a challenge to precisely construct complex supramolecular structures through the one-step method.

Nature has given us the enlightenment in the exploration of controllable self-assembly of supramolecules with the self-sorting process, such as the complementary pairing of DNA and the folding of polypeptides of proteins[28]. In recent years, much more attention has been paid to the design of different ligands for the diversity and controllability of supramolecular assembly through social self-discrimination or narcissistic self-recognition manner[29–33]. Particularly, social self-discrimination is relatively common for tpy moieties, and multiple different building blocks can be engineered to form impressively complex structures[34,35]. For example, phenanthroline with modification at 2,9-positions and tpy with modification at 6,6″-position could show the complete social self-sorting behaviors when mixed with tpy (Fig. 1a)[36,37]. In a recent review, Schmittel summarized a dynamic library formed by hetero-ligand motifs and introduced a lot of elegant structures constructed by the social self-sorting process[38]. In contrast, the narcissistic self-recognition manner for tpy moieties is rarely reported[39–41]. It has been more than 20 years since Lehn reported the first narcissistic self-sorting of a double helix, and there have also been many reports about narcissistic self-sorting in metal coordination[42–46]. However, most of these studies considering the narcissistic self-sorting of the structure as a whole, while the report focusing on narcissistic self-sorting used as self-recognition sites remains limited[45–51]. Furthermore, the homogeneous interactions in biological systems also prompted scientists to urgently develop ligand motifs for self-recognition[52,53].

In this paper, to achieve coordination with high selectivity and specificity, we design two ortho-modified terpyridine ligands, i.e., MA and MB as model systems to form head-to-tail coordination complexes $Zn_2(MA)_2$ and $Zn_3(MB)_2$, respectively (Fig. 1b). The single-crystal structures of $Zn_2(MA)_2$ and $Zn_3(MB)_2$ show the non-coaxial feature of ortho-modified tpy after coordination, which is apparently different from the rotation of conventional tpy around its axis[19,54]. Remarkably, MA, MB, and tpy mixture with Zn(II) could assemble into three distinct complexes

$Zn_2(MA)_2$, $Zn_3(MB)_2$, and $Zn(tpy)_2$, suggesting the narcissistic self-sorting behaviors. We then combined these moieties with tpy to form sterically congested multitopic ligands (LA and LB) for precise self-assembly of hexagonal macrocycles $Zn_9(LA)_6$ and $Zn_{12}(LB)_6$ (Fig. 1c). It's a common strategy to promote the hierarchical assembly of metallacycles by introducing orthogonal interactions[55–57]. The tpy-based metallacycles could hierarchically self-assembly into hollow tubular or berry-type nanostructure through the face-to-face stacking according to the previous reports[15,21,58,59]. Interestingly, it is found that the significantly steric congestion and non-coaxial structure of $Zn_9(LA)_6$ caused the restriction in the edging rotation of the hexagons, which further causes the hexagons to hierarchically assemble into giant cyclic nanostructures and metallogels via edge-to-edge stacking rather than face-to-face stacking.

## Results and discussion

**Synthesis and characterization of model system MA and complex $Zn_2(MA)_2$.** In our design, the modification at terpyridyl 6-position played a central role in the self-soring assembly. The simple and efficient synthesis process of MA only included a three-step reaction of the starting material, followed by the one-step purification of the product using column chromatography. Especially, the key compound **1** was synthesized by a typical Kröhnke reaction with pyridinium salt **5** (Supplementary Fig. 1)[60]. The final motif MA was obtained via Suzuki coupling reaction in a good yield (77%) and characterized by NMR and mass spectrometry (Supplementary Figs. 2–8). After that, MA and $Zn(NO_3)_2 \cdot 6H_2O$ (with a precise stoichiometric ratio of 1:1, Fig. 2a) were mixed in $CHCl_3$ and MeOH (1:3, v/v) at 50 °C for 12 h, followed by the addition of excessive $NH_4PF_6$ (to exchange $NO_3^-$ with $PF_6^-$) in methanol to give a white precipitate $Zn_2(MA)_2$ in a yield of 91%.

$^1H$ NMR spectra of MA and complex $Zn_2(MA)_2$ were shown in Fig. 2b. Given the dissymmetrical nature of MA, five kinds of expected pyridines signals attributed to the terpyridine moieties were confirmed by 2D-COSY results (Supplementary Fig. 6). In the spectrum of complex $Zn_2(MA)_2$, the peaks were sharp and well-resolved. The 2D-COSY and NOESY for $Zn_2(MA)_2$ (Supplementary Figs. 9–14) also showed five sets of pyridines signals as MA, suggesting a highly symmetrical structure of complex $Zn_2(MA)_2$. Compared with MA, 6-position (6 and a6) signals of tpy were shifted upfield due to the electron shielding effect[61]. At the same time, the proton signals of e-tpy, f-Ph, and g-Ph were also shifted to upfield, which should be attributed to the existence of π–π interaction between tpy moiety and phenyl in the other ligand. In ESI-MS (Supplementary Fig. 15A), one prominent set of peaks with charge states from 2+ to 4+ was observed (due to the loss of different numbers of $PF_6^-$). After deconvolution, the molecular weight of complex $Zn_2(MA)_2$ was 1940 Da, matching well with its expected chemical composition of two MA moieties, two Zn(II) ions, and four $PF_6^-$. All the experimental isotope patterns agree excellently with the corresponding simulated isotope patterns (Supplementary Fig. 16). In traveling wave ion mobility-mass spectrometry (TWIM-MS)[23], complex $Zn_2(MA)_2$ showed a series of charge states with narrow drift time distribution ranging from 2+ to 4+, indicating the formation of a discrete product but without any isomers and conformers (Supplementary Fig. 15B).

To further confirm the structure, the single crystal of $Zn_2(MA)_2$ was obtained by slowly diffusing the vapor of ethyl acetate into $Zn_2(MA)_2$ in acetonitrile for over 2 weeks. As expected, X-ray crystallographic analysis (Fig. 2c, d and Supplementary Table 1) showed that two Zn(II) are sandwiched between two MA to form a dimer with the head-to-tail structure

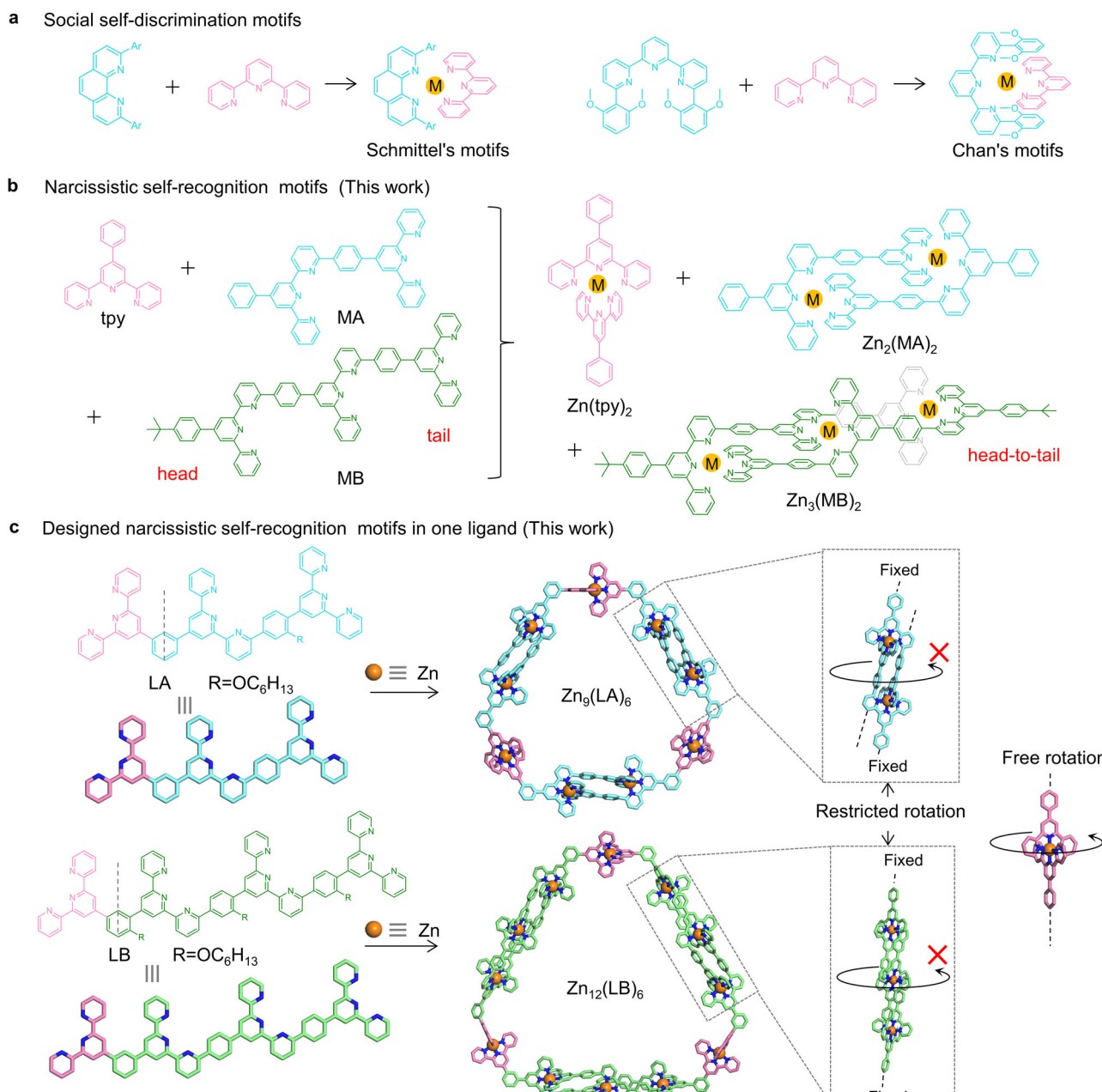

**Fig. 1 The self-assembly of metallo-macrocycles by the narcissistic self-recognition tpy moieties.** Two different self-sorting processes in coordination-driven self-assembly; **a** social self-discrimination motifs in literatures; **b** narcissistic self-recognition motifs in the current work. We combined the narcissistic self-recognition tpy moieties with tpy to form multitopic ligands for precise self-assembly of metallo-macrocycles. **c** Schematics for the self-assembly of hexagonal macrocycles $Zn_9(LA)_6$ and $Zn_{12}(LB)_6$.

(Supplementary Data 1 and Supplementary Movie 1). The phenyl groups in the middle are parallel to each other (yellow part), and the distance between the two interlayers is 3.57 Å, indicating the existence of π–π interactions[41,62].

**Synthesis and characterization of model system MB and complex $Zn_3(MB)_2$.** By referring to the synthetic and characterization procedures of MA, we further designed and obtain another model system MB (Fig. 3a and Supplementary Fig. 17), which includes one additional tpy unit compared to MA and thereby exhibits an enhanced sterically congested effect. An extra *tert*-butyl was introduced to improve the solubility. Tpy-based molecules have been extensively studied since the 1930s, however, it is worth noting that the unique structure of MB, has never been

reported[40,63,64]. The synthesis of MB (yield 75%) and self-assembly of complex $Zn_3(MB)_2$ (yield 87%) (MB:Zn(II) = 2:3) followed the same procedure as that of MA and $Zn_2(MA)_2$.

The [1]H NMR data of MB and $Zn_3(MB)_2$ were shown in Fig. 3b. In the spectrum of MB with three tpy units, eight kinds of pyridines signals could be observed (Supplementary Figs. 18–24). Moreover, the complex $Zn_3(MB)_2$ exhibited a more complicated structure and spectrum than $Zn_2(MA)_2$ (Supplementary Figs. 25–30), due to the enhanced steric hindrance effect caused by the introduction of the additional tpy unit. For each tpy, the differences in chemical shift (Δδ) were 1.48 ppm and 0.8 ppm for 6 position proton and a6 proton, respectively, suggesting that the 6-position of a-tpy shows the strong shielding effect. The signals of tpy-3 and tpy-5 displayed upfield or downfield shifts after

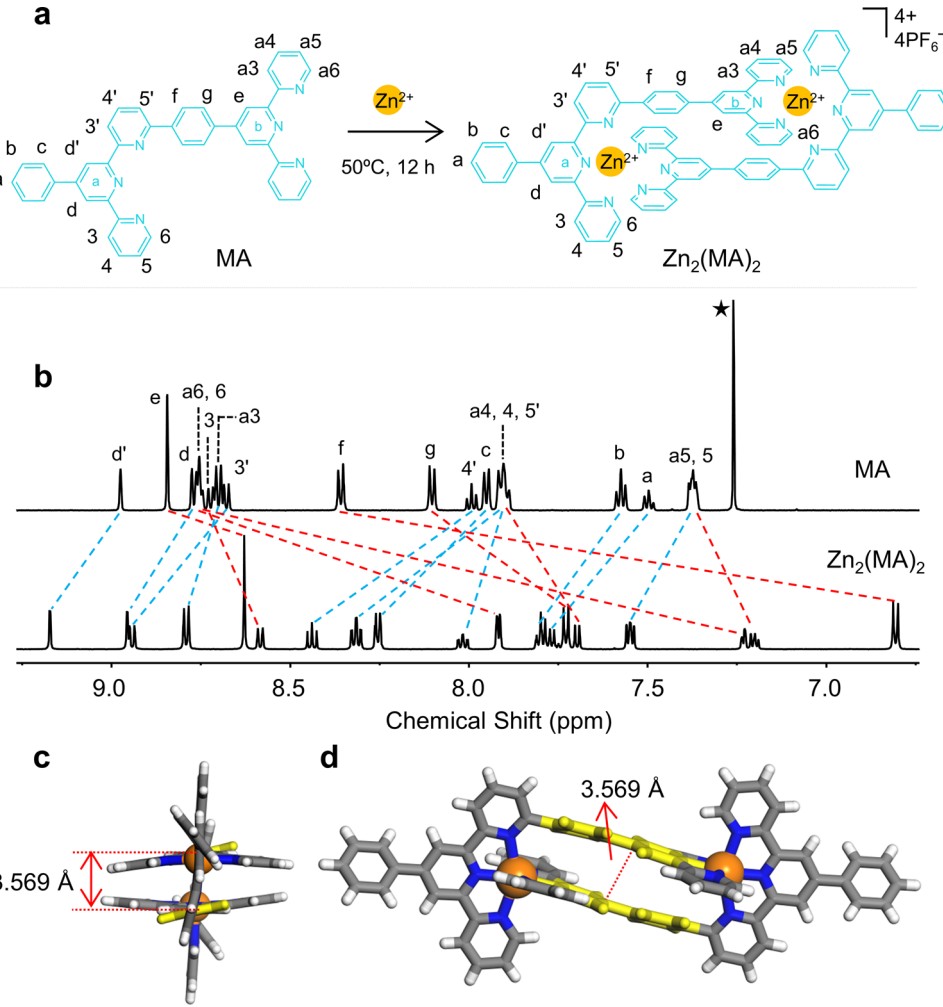

**Fig. 2 The self-assembly and characterization of head-to-tail complex Zn₂(MA)₂. a** The self-assembly of model system MA with Zn(II). **b** The ¹H NMR spectra (600 MHz, 300 K) of MA in CDCl₃ and complex Zn₂(MA)₂ in CD₃CN (3 mg/mL). **c** X-ray crystal structure in a side view of complex Zn₂(MA)₂. **d** X-ray crystal structure in a front view of complex Zn₂(MA)₂. Non-coordinated anions and solvent are omitted for clarity (C, gray or yellow; H, white; N, blue; Zn, orange).

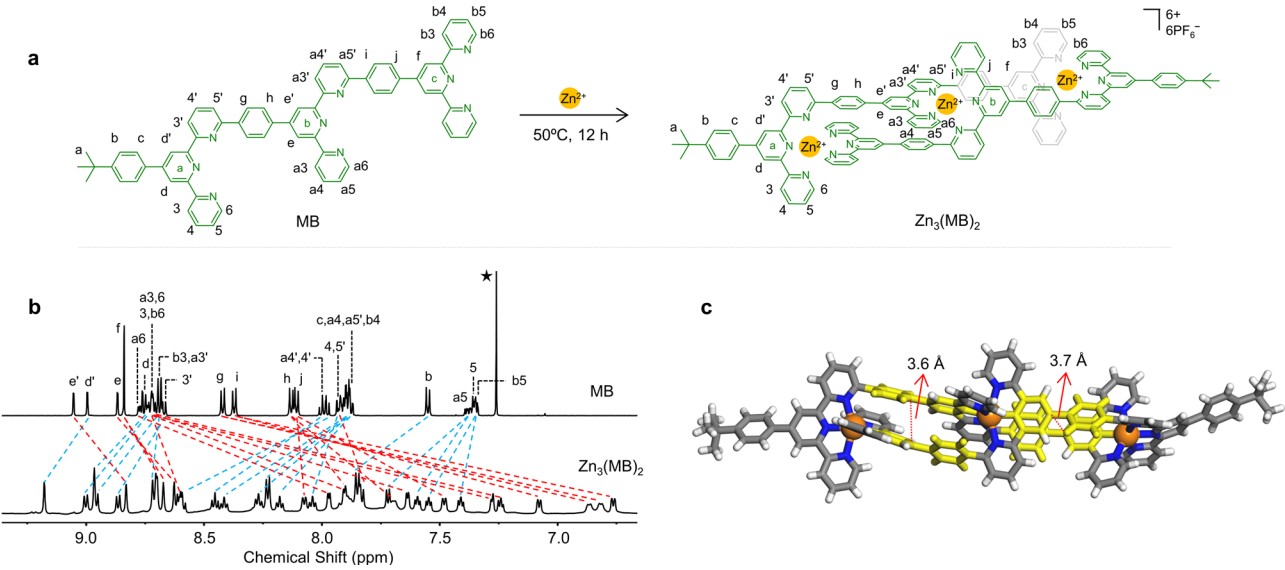

**Fig. 3 The self-assembly and characterization of head-to-tail complex Zn₃(MB)₂. a** The self-assembly of model system MB with Zn(II). **b** The ¹H NMR spectra (600 MHz, 300 K) of MB in CDCl₃ and Zn₃(MB)₂ in CD₃CN (3 mg/mL). **c** X-ray crystal structure in a front view of Zn₃(MB)₂. Non-coordinated anions and solvent are omitted for clarity (C, gray or yellow; H, white; N, blue; Zn, orange).

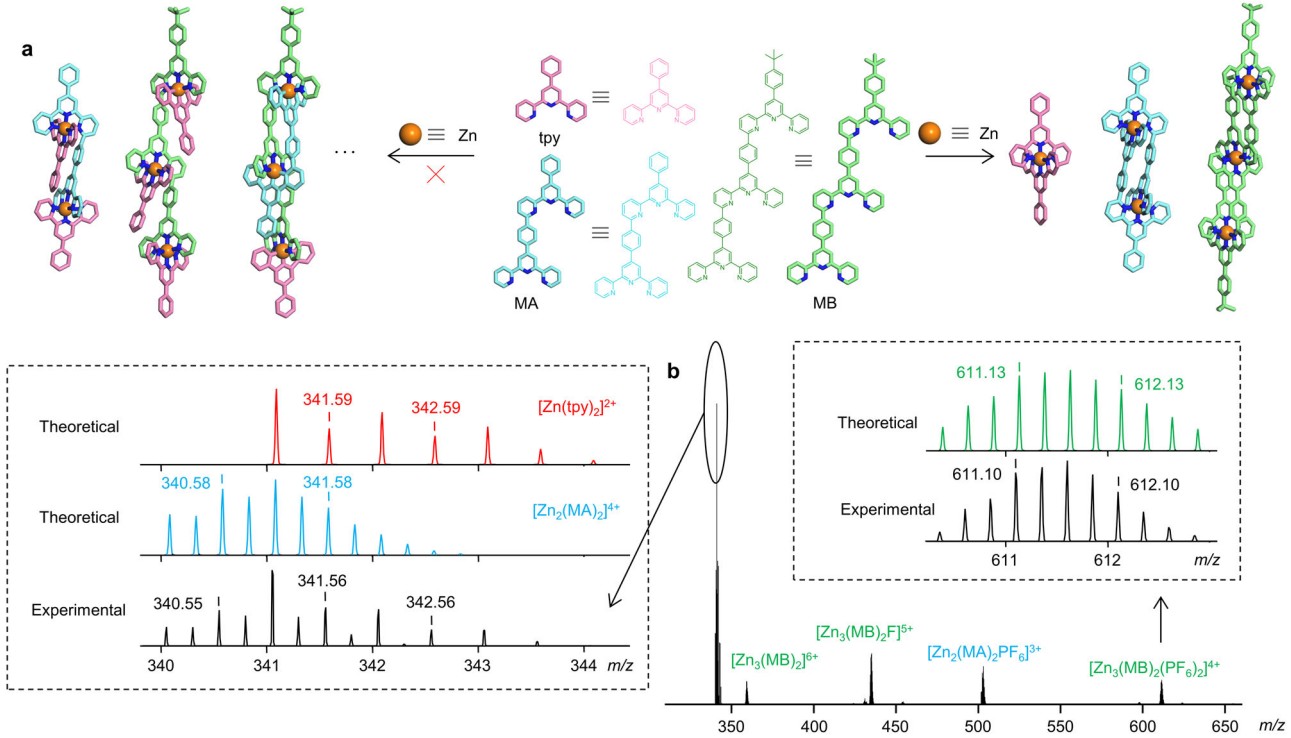

**Fig. 4 The narcissistic self-sorting behavior of MA, MB, and tpy. a** A schematic representation for the narcissistic self-sorting process. **b** ESI-MS of the assembly of mixed MA, MB, and tpy in an equimolar ratio with the corresponding amount of Zn(II).

coordination, which were consistent with the previous studies[65]. In addition, the restrictive structure induced the split of NMR signals. Both the signals of the middle phenyls and tpy-b6 split into two sets of peaks, which indicate their rotation restriction caused by the altered chemical environment. ESI-MS and TWIM-MS (Supplementary Figs. 31, 32) spectra of $Zn_3(MB)_2$ also exhibited a similar prominent set of peaks with different charge states and narrow drift time distribution ranging from 2+ to 4+, suggesting the formation of a single product without any overlapping isomers or conformers. The molecular weight of complex $Zn_3(MB)_2$ (3020 Da) agrees well with its expected chemical composition of two MB ligands, three Zn(II) ions, and six $PF_6^-$.

The single-crystal data of $Zn_3(MB)_2$ was obtained by slowly diffusing the vapor of carbon tetrachloride into acetonitrile solution for over 3 weeks. Complex $Zn_3(MB)_2$ also showed a sandwich-like structure with a certain degree of distortion, suggesting higher steric congestion caused the helical shape (Fig. 3c). The different distances between the two middle phenyls (yellow part) were 3.6 and 3.7 Å, respectively, which confirmed the helical structure (Supplementary Data 2, Supplementary Table 2, and Supplementary Movie 2). More importantly, under the sterically congested environment, the different tpy units coordinated with the same Zn(II) cannot maintain the vertical alignments and are forced to exhibit twisted structures. Furthermore, $Zn_3(MB)_2$ has two chiral conformations due to the unique head-to-tail coordination mode, and both of the conformations could be found in a single crystal (Supplementary Fig. 33).

**Self-sorting behavior of MA, MB, and tpy.** To evaluate the selectivity of these model systems, we investigated the self-sorting behavior of MA, MB, and conventional tpy[66]. In the self-sorting study, MA, MB, and conventional tpy were mixed together in an equimolar ratio with the corresponding amount of Zn(II) for the

overnight self-assembly at 50 °C. ESI-MS clearly illustrated three sets of signals for the corresponding complexes but without statistical complexes (Fig. 4a, b), indicating that these model complexes exhibit ideal self-sorting properties in a pluralistic system. Both ESI-MS and NMR of any binary mixture (Supplementary Figs. 34–36) showed independent signals for three complexes $Zn_2(MA)_2$, $Zn_3(MB)_2$, and $Zn(tpy)_2$, suggesting a characteristic narcissistic self-sorting[29,67,68]. Moreover, we also monitored the kinetic process of narcissistic self-sorting, and the system basically reached equilibrium after 24 h (Supplementary Fig. 37). As expectation, in accord to the maximum occupancy of coordination sites proposed by Lehn[46], MA, MB, and tpy tend to narcissistically coordinate to form the most energy favorable structure. In this context, we are inspired to use MA and MB as self-recognition sites to further construct complex supramolecular structures.

**Synthesis and self-assembly of ligands LA and LB with multiple tpy moieties.** It is reported that the ditopic tpy ligands with 120° angle tend to self-assemble into a mixture of macrocycles with uncontrollable size and structure[20]. However, we expect that the dissymmetrical tritopic (LA) and tetratopic (LB) (Fig. 1c) ligands with 120° angle that contain narcissistic self-sorting moieties could be used to precisely construct complex supramolecular macrocycles $Zn_9(LA)_6$ and $Zn_{12}(LB)_6$ (Supplementary Movie 3 and Supplementary Movie 4). In order to improve the solubility of ligands, alkoxy chains (R = -OC$_6$H$_{13}$) were introduced. Compared with LA (yield, 76%), the synthesis of LB with one more tpy units turns to be significantly challenging (Supplementary Fig. 38 and Supplementary Fig. 39). Fortunately, we successfully obtained LB (yield, 37%) by adding CH$_3$COOH/DMF as an auxiliary agent. The obtained ligands LA and LB were assembled into complexes $Zn_9(LA)_6$ (yield, 86%) and $Zn_{12}(LB)_6$ (yield, 84%) and were characterized by NMR, MALDI-TOF, ESI-MS, and TWIM-MS (Fig. 5 and Supplementary Figs. 40–70).

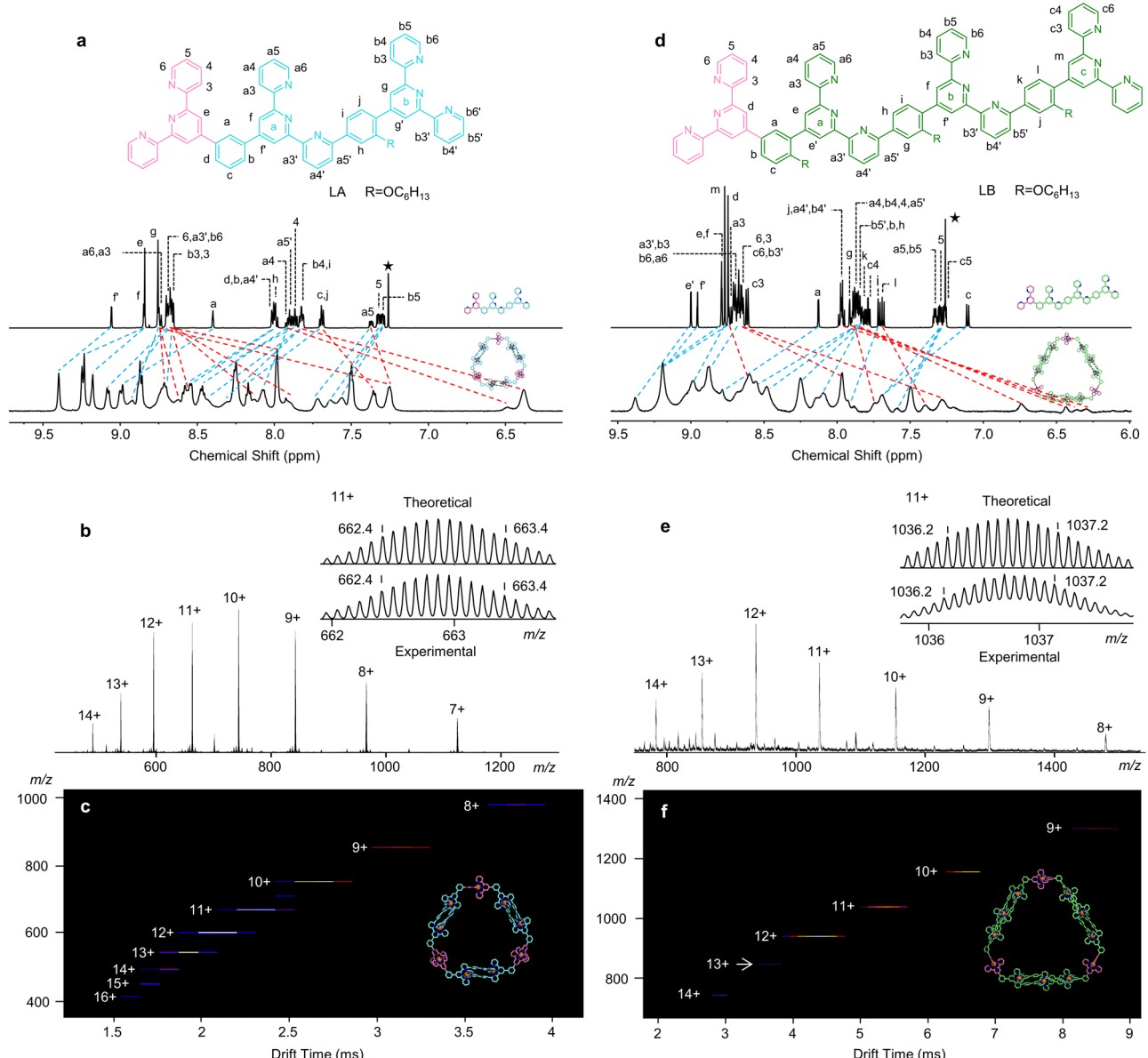

**Fig. 5 Characterization of the hexagonal macrocycles Zn₉(LA)₆ and Zn₁₂(LB)₆. a** $^1$H NMR spectra (600 MHz, 300 K) of ligand LA in CDCl₃ and hexagon Zn₉(LA)₆ in CD₃CN (4 mg/mL). **b** ESI-MS, and **c** TWIM-MS plots ($m/z$ vs drift time) of Zn₉(LA)₆. **d** $^1$H NMR spectra (600 MHz, 300 K) of ligand LB in CDCl₃ and hexagon Zn₁₂(LB)₆ in CD₃CN (4 mg/mL). **e** ESI-MS and **f** TWIM-MS plots ($m/z$ vs drift time) of Zn₁₂(LB)₆.

Compared with LA and LB, $^1$H NMR spectra of the complexes Zn₉(LA)₆ and Zn₁₂(LB)₆ were significantly broadened (Fig. 5a, d), suggesting the assembly of large complexes with a slower tumbling on the NMR time scale[24]. Importantly, the peaks of pyridines on the b-tpy and c-tpy turn broader, due to the steric congestion of the coordination units and the restriction in free rotation of the phenyl caused by the high steric congestion of alkoxy chains. In the spectrum of LA, seven kinds of expected pyridines signals were observed with respect to the tpy moieties. Owing to the continuous dissymmetrical modification of a- and b-tpy, LB displayed more complicated proton signals (i.e., ten distinguished kinds of pyridine peaks) than that of LA. All six positions of pyridines were shifted towards upfield, because of the electron shielding effects after complexation. However, the chemical shifts of different protons differ significantly owing to the variable shielding strength. For instance, the proton at a6 position showed a maximum chemical shift of $\Delta\delta = 1.41$ ppm; while a slightly upfield-shift $\Delta\delta = 0.55$ ppm was observed for b6

position. 2D-COSY and NOESY for two complexes Zn₉(LA)₆ and Zn₁₂(LB)₆ (Supplementary Figs. 53–56 and Supplementary Figs. 59–62) also showed the same sets of tpy signals as ligands, suggesting the symmetrical structures of these two complexes. The diffusion-ordered NMR spectroscopy (DOSY) provided dimensional information for Zn₉(LA)₆ and Zn₁₂(LB)₆. As shown in Supplementary Fig. 66, all proton signals appeared at the same band, indicating the formation of discrete assemblies. The diffusion coefficients in CD₃CN were log D = −9.61 for Zn₉(LA)₆ and log D = −9.64 for Zn₁₂(LB)₆, respectively. The experimental hydrodynamic radii (rH) for Zn₉(LA)₆ (2.4 nm) and Zn₁₂(LB)₆ (2.6 nm) agreed well with the modeling structures.

In ESI-MS (Fig. 5b, e), one prominent set of peaks with charge states (from 7+ to 14+ for Zn₉(LA)₆ and from 8+ to 14+ for Zn₁₂(LB)₆) was observed on account of the loss of different numbers of PF₆⁻. After deconvolution, the molecular weights of these two complexes were 8869 and 12974 Da, respectively, matching well with the expected chemical composition of

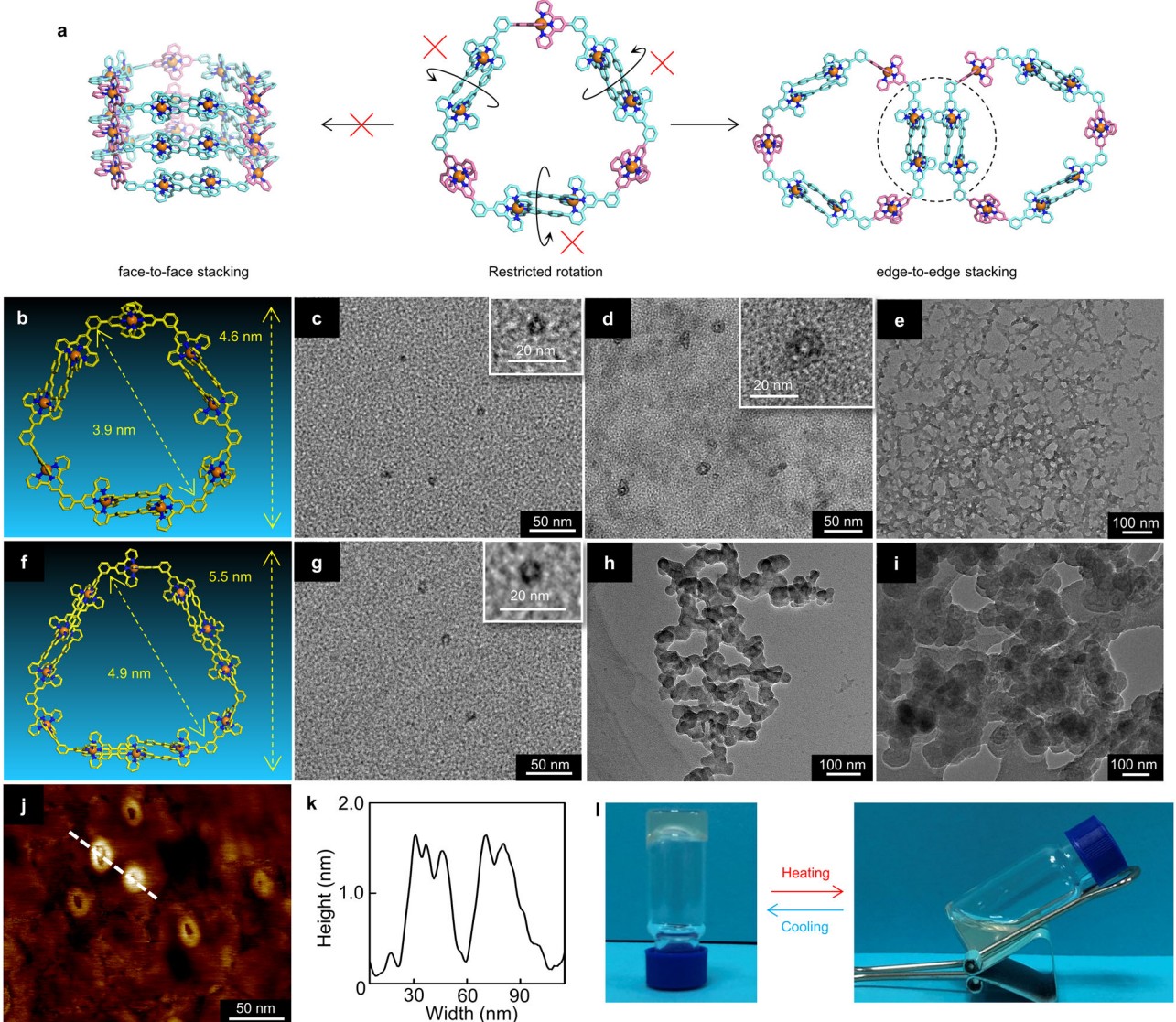

**Fig. 6 Characterization of the hierarchical self-assembly of Zn$_9$(LA)$_6$ and Zn$_{12}$(LB)$_6$. a** Schematic diagram of edge-to-edge stacking by Zn$_9$(LA)$_6$. **b** The energy-minimized structure of Zn$_9$(LA)$_6$ from molecular modeling. **c** TEM images of individual Zn$_9$(LA)$_6$ at a concentration of $10^{-6}$ M in CH$_3$CN (scale bar, 50 and 20 nm for zoom-in image). **d** TEM images of the assembled cyclic nanostructures of Zn$_9$(LA)$_6$ at a concentration of $10^{-5}$ M in CH$_3$CN (scale bar, 50 and 20 nm for zoom-in image). **e** TEM images of the 3D network structure formed by assembled cyclic nanostructures of Zn$_9$(LA)$_6$ at a concentration of $10^{-4}$ M in CH$_3$CN (scale bar, 100 nm). **f** The energy-minimized structure of Zn$_{12}$(LB)$_6$ from molecular modeling. **g** TEM images of individual Zn$_{12}$(LB)$_6$ at a concentration of $10^{-6}$ M in CH$_3$CN (scale bar, 50 and 20 nm for zoom-in image). **h** TEM images of the assembled cyclic nanostructures of Zn$_{12}$(LB)$_6$ at a concentration of $10^{-5}$ M in CH$_3$CN (scale bar, 100 nm). **i** TEM images of the 3D network structure formed by assembled cyclic nanostructures of Zn$_{12}$(LB)$_6$ at a concentration of $10^{-4}$ M in CH$_3$CN (scale bar, 100 nm). **j** AFM images of the hierarchically assembled cyclic nanostructures by Zn$_9$(LA)$_6$ (scale bar, 50 nm). **k** Cross-section of the cyclic nanostructure shown in image **j**. **l** Photograph of Zn$_{12}$(LB)$_6$ gel formation at a concentration of 25 mg/mL in CH$_3$CN.

Zn$_9$(LA)$_6$ (6 LA ligands, 9 Zn(II) ions, and 18 PF$_6^-$) and Zn$_{12}$(LB)$_6$ (6 LB ligands, 12 Zn(II) ions, and 24 PF$_6^-$). All the experimental isotope patterns agreed excellently with the corresponding simulated isotope patterns (Supplementary Figs. 69, 70). Moreover, the TWIM-MS spectra of complexes Zn$_9$(LA)$_6$ and Zn$_{12}$(LB)$_6$ also showed narrow drift time distribution, supporting the formation of single species with rigid structures (Fig. 5c, f). The full characterization of NMR, ESI-MS, and TWIM-MS confirmed that the two types of tpy moieties in ligands LA and LB could precisely self-assemble into hexagonal macrocycles through the narcissistic self-recognition mechanism. Such moieties with high coordination selectivity and specificity may find their potential in the constructions of more complex

metallo-supramolecules with precisely controlled shapes and sizes.

Interestingly, when complexes Zn$_9$(LA)$_6$ and Zn$_{12}$(LB)$_6$ were mixed in an equimolar ratio at 50 °C for overnight, ESI-MS and TWIM-MS clearly illustrated two additional sets of signals besides Zn$_9$(LA)$_6$ and Zn$_{12}$(LB)$_6$ (Supplementary Fig. 71). The compositions of Zn$_{10}$(LA)$_4$(LB)$_2$ and Zn$_{11}$(LA)$_2$(LB)$_4$ were confirmed by analyzing the ESI-MS and TWIM-MS data (Supplementary Figs. 72, 73). No exchange for odd number ligands was observed in this mixture (Supplementary Figs. 74–77), suggesting that it could be a subcomponent exchange based on the dominant coordination dimer rather than the usual ligand exchange[69,70]. It indicated that the dimer structures based on

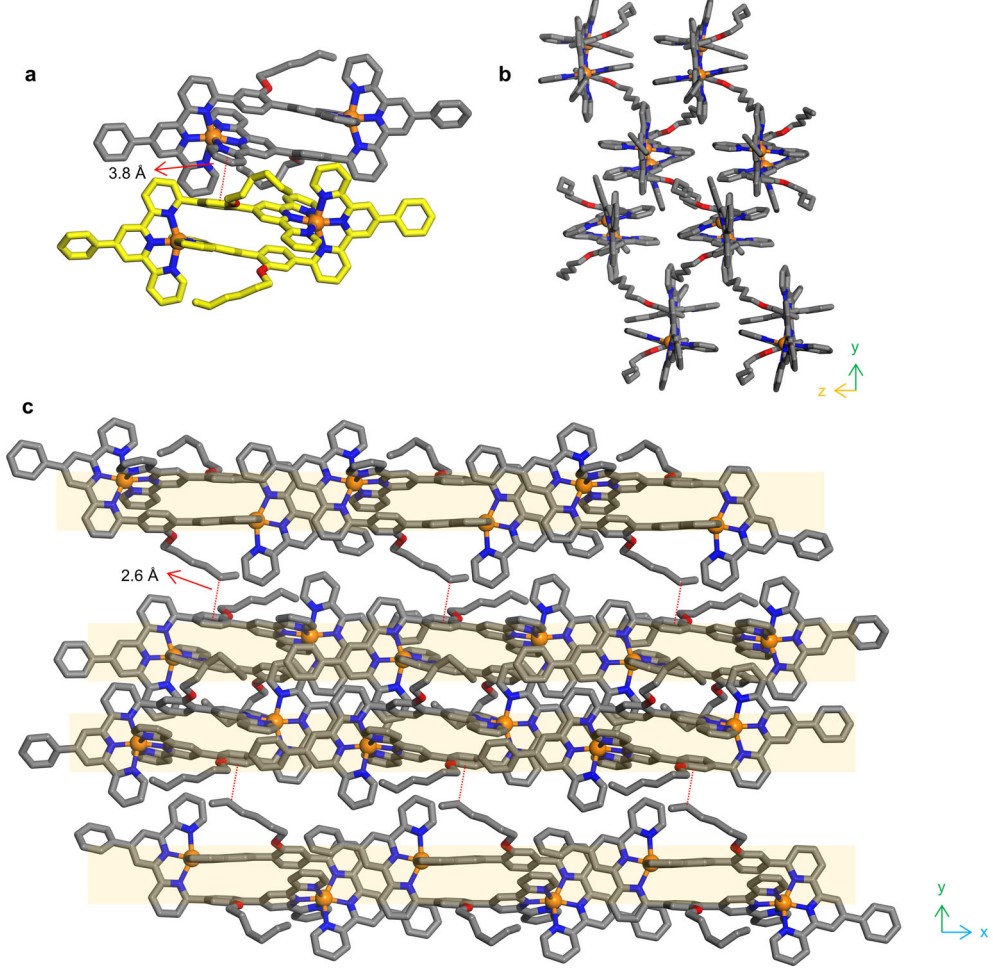

**Fig. 7 Crystallographic structure of complex Zn$_2$(MA-OC$_6$H$_{13}$)$_2$. a** Crystal packing of two complexes Zn$_2$(MA-OC$_6$H$_{13}$)$_2$. **b** Crystal packing structure in a side view of complex Zn$_2$(MA-OC$_6$H$_{13}$)$_2$. **c** Crystal packing structure in a front view of complex Zn$_2$(MA-OC$_6$H$_{13}$)$_2$. H atoms, non-coordinated anions, and solvent are omitted for clarity (C, gray or yellow; N, blue; Zn, orange).

ditopic and tritopic tpy moieties not only exhibit higher selectivity but also show higher stability than single-tpy unit.

**Hierarchical self-assembly**. Our study shows that, with increasing the concentration of Zn$_9$(LA)$_6$ and Zn$_{12}$(LB)$_6$ in acetonitrile at room temperature, both of the solutions could form metallogels at a concentration of 45 mg/mL for Zn$_9$(LA)$_6$ and 25 mg/mL for Zn$_{12}$(LB)$_6$ (Fig. 6l). The gels were also temperature-responsive and could be converted into solutions after heating at 50 °C for 1 h. To explore the mechanism of gelation, we then used TEM to investigate the morphology of Zn$_9$(LA)$_6$ and Zn$_{12}$(LB)$_6$ at different concentrations (Supplementary Figs. 78, 79). At low concentrations (10$^{-6}$ M), the assemblies were almost distributed in a monodispersed manner. Both complexes exhibited unique hollow structures, suggesting high rigidity, stability, and large inner space. The measured sizes of the two complexes were 5.5 nm for Zn$_9$(LA)$_6$ and 6.5 nm for Zn$_{12}$(LB)$_6$, respectively (Fig. 6b, c, f, g). However, at a high concentration (10$^{-5}$ M), some giant cyclic nanostructures in larger size were observed (Fig. 6d). The diameters of these structures were much larger than that of individual Zn$_9$(LA)$_6$, implying the clustering of Zn$_9$(LA)$_6$. The height of the hierarchically assembled nanostructures measured by the AFM image (Fig. 6j, k and Supplementary Figs. 80–83) was about 1.5 nm, which was very close to the height of the monolayer. When the concentration further increased, these giant macrocycles formed a three-dimensional network via

intermolecular interaction (Fig. 6e). For Zn$_{12}$(LB)$_6$ with a larger size, although similar monolayer nanostructures were observed (Fig. 6h), the degree of aggregation was significantly increased (Supplementary Figs. 84–86). Further, at a much higher concentration of 10$^{-4}$ M, the formed giant macrocycles showed obvious vertical stacking in space (Fig. 6i).

As well-known, the <tpy-M(II)-tpy> unit could rotate freely around the axis without space obstruction. However, due to the special spatially complementary coordination of LA and LB, these two axial dissymmetrical ligands exhibited entirely steric congestions. Therefore, it can be inferred that the hexagons are basically composed of vertical edges, since the rotation of the dimer can result in the disassembly (Fig. 6a). In order to study the edge-to-edge stacking of Zn$_9$(LA)$_6$, we tried to grow single crystal of Zn$_9$(LA)$_6$. However, all efforts to grow a single crystal of Zn$_9$(LA)$_6$ has proven to be unsuccessful. Instead, we synthesized a model ligand MA-OC$_6$H$_{13}$ (Supplementary Figs. 87–94) and further got the complex Zn$_2$(MA-OC$_6$H$_{13}$)$_2$ which has exactly the same structure as the ditopic part of Zn$_9$(LA)$_6$ (Supplementary Figs. 95–102). The single-crystal packing of Zn$_2$(MA-OC$_6$H$_{13}$)$_2$ provided strong evidence for the edge-to-edge stacking of Zn$_9$(LA)$_6$ (Supplementary Data 3 and Supplementary Table 3). As shown in Fig. 7, the intermolecular packing of Zn$_2$(MA-OC$_6$H$_{13}$)$_2$ is more compact than that of Zn$_2$(MA)$_2$ (Supplementary Fig. 103), which indicated that the introduction of alkyl chains can enhance the intermolecular interactions. The distance

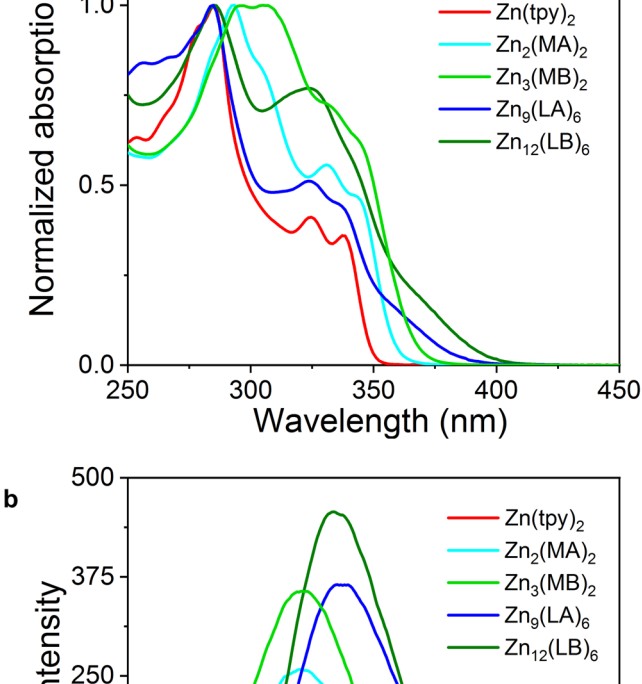

**Fig. 8 Photophysical data of the complexes. a** Normalized UV/Vis absorption of all complexes in $CH_3CN$. **b** PL spectra of all complexes in $CH_3CN$ ($10^{-6}$ M, $\lambda_e = 325$ nm).

between pyridine and central phenyl of another molecule is 3.8 Å, which indicated the existence of $\pi$–$\pi$ interaction. The distance between the alkyl chain and central phenyl of another molecule is 2.6 Å, which indicated the existence of CH–$\pi$ interactions. Since $Zn_{12}(LB)_6$ could provide additional sites in the vertical direction than complex $Zn_9(LA)_6$, $Zn_{12}(LB)_6$ is hard to find the discrete cyclic nanostructures as that of $Zn_9(LA)_6$.

**Photophysical properties.** Considering the enhanced conjugation and restricted rotation of the structures, we further investigated UV/Vis absorption and photoluminescence of all ligands and complexes in the solution state ($10^{-6}$ M). As shown in Fig. 8a, compared with the absorption spectra of ligands (Supplementary Figs. 104, 105), all complexes displayed a significant redshift originating from intra-ligand charge transfer (ILCT)[71]. In sharp contrast with $Zn(tpy)_2$, the maximum emission wavelength of $Zn_2(MA)_2$ displayed a ca. 75 nm redshift, ascribed to the enhanced conjugation (Fig. 8b). However, complex $Zn_3(MB)_2$ with a more complicated structure also showed a similar maximum emission wavelength to $Zn_2(MA)_2$, the reason should be the twisted structures induced the conjugation. For hexagonal macrocycles $Zn_9(LA)_6$ and $Zn_{12}(LB)_6$, the restrictive intramolecular rotation in the congested and non-coaxial structures caused further enhance of their fluorescence intensity. Moreover, due to the introduction of alkoxy chains as electron donors, the

maximum emission peaks of $Zn_9(LA)_6$ and $Zn_{12}(LB)_6$ exhibited 20 nm red-shifted than that of $Zn_2(MA)_2$ and $Zn_3(MB)_2$.

## Conclusion
In summary, we have designed and synthesized a series of multitopic tpy ligands with designed coordination selectivity by ortho-modification. As shown by the single-crystal detections, the model systems MA and MB can assemble into head-to-tail coordination complexes $Zn_2(MA)_2$ and $Zn_3(MB)_2$, respectively. Moreover, the assembly of mixtures of MA, MB, and tpy exhibit excellent selectivity due to the narcissistic self-sorting mechanism. Further, we introduced these moieties with high selectivity into multitopic ligands LA and LB, which were used to further precisely construct the hexagonal metallo-supramolecules. Such a design caused the restriction in rotations of the hexagons, which further leads to their hierarchical assembly into giant cyclic nanostructures and metallogels. Moreover, these complexes showed significantly enhanced fluorescence intensity in the solution state than the complexes based on conventional tpy due to the additional ligand conjugation and the restricted chemical environment. Our design and fabrication of dissymmetrical coordination moieties could pave a new avenue for the development of a set of congested coordination pairs with high selectivity and specificity for the assembly of sequence-specific metallo-supramolecular architectures.

## Methods
**General procedures.** All reagents were purchased from Sigma-Aldrich, Matrix Scientific, Alfa Aesar, Jilin Chinese Academy of Sciences—Yanshen Technology Co. Ltd., and used without further purification. Column chromatography was conducted using $SiO_2$ (VWR, 40–60 μm, 60 Å) and the separated products were visualized by UV light.

**Synthesis.** All the new compounds were fully characterized and spectra are given in Supplementary Methods.

**Nuclear magnetic resonance (NMR).** NMR spectra data were recorded on a 400, 500, and 600 MHz Bruker Avance NMR spectrometer in $CDCl_3$ or $CD_3CN$ with TMS as reference. For NMR spectra see Supplementary Information.

**ESI-MS and TWIM-MS.** Electrospray ionization (ESI) mass spectra were recorded with a Waters Synapt G2 tandem mass spectrometer, using solutions of 0.5 mg sample in 1 mL of MeCN/MeOH (3:1, v/v) for complexes. The TWIM-MS experiments were performed under the following conditions: ESI capillary voltage, 3 kV; sample cone voltage, 30 V; extraction cone voltage, 3.5 V; source temperature 100 °C; desolvation temperature, 100 °C; cone gas flow, 10 L/h; desolvation gas flow, 700 L/h ($N_2$); source gas control, 0 mL/min; trap gas control, 2 mL/min; helium cell gas control, 100 mL/min; ion mobility (IM) cell gas control, 30 mL/min; sample flow rate, 5 μL/min; IM traveling wave height, 25 V; and IM traveling wave velocity, 1000 m/s.

**Single-crystal X-ray diffractions.** X-ray diffraction data for $Zn_2(MA)_2$ and $Zn_2(MA-OC_6H_{13})_2$ were measured by a Bruker D8 Venture X-ray single-crystal diffractometer using a Cu·Kα radiation ($\lambda = 1.54178$ Å) at 100 K. X-ray diffraction data for $Zn_3(MB)_2$ was collected using synchrotron radiation and MAR325 CCD detector at Shanghai Synchrotron Radiation BL17B Beamline. Crystallographic data and structural characteristics of $Zn_2(MA)_2$, $Zn_2(MA-OC_6H_{13})_2$, and $Zn_3(MB)_2$ are summarized in Supplementary Tables 1–3, respectively. Crystallographic information files for the complexes are provided in Supplementary Data 1–3, respectively.

**Transmission electron microscopy (TEM) analysis.** The sample of $Zn_9(LA)_6$ and $Zn_{12}(LB)_6$ were dissolved in $CH_3CN$ at concentrations of $10^{-6}$, $10^{-5}$, and $10^{-4}$ M, respectively. The solutions were left overnight at 298 K and then dropped cast onto copper grids (ultrathin carbon supported by a lacey carbon film on a 400 Mesh copper grid) and the extra solution was absorbed by filter paper to avoid aggregation. The TEM images of the drop cast samples were taken with a JEM-2100F transmission electron microscope.

**AFM imaging.** AFM imaging was performed on a Bruker Dimension Icon AFM system with ScanAsyst and the data was processed by NanoScope Analysis version 2.0 (Bruker Software, Inc.). The sample of $Zn_9(LA)_6$ and $Zn_{12}(LB)_6$ were dissolved

in $CH_3CN$ at a concentration of $10^{-5}$ M, The solution was left overnight and then dropped cast onto a silicon wafer after surface cleaning.

**Photophysical measurements**. UV-vis spectra of solutions were recorded on a PerkinElmer LAMBDA-365 spectrophotometer. Fluorescence emission spectra were measured by a Shimadzu spectrofluorimeter RF-5301PC. Solutions were placed in 1 cm path length quartz cells.

**Molecular modeling**. Energy-minimized structures were obtained following the settings in the literature. Calculations were proceeded with Geometry Optimization and followed by Anneal in Forcite module of Materials Studio version 8.0 program (Accelrys Software, Inc.).

## Data availability

The authors declare that all data supporting the findings of this study are available within the article and Supplementary Information files, and also from the corresponding author upon reasonable request. The X-ray crystallographic coordinates for $Zn_2(MA)_2$ (Supplementary Data 1), $Zn_3(MB)_2$ (Supplementary Data 2), and $Zn_2(MA-OC_6H_{13})_2$ (Supplementary Data 3) have been deposited at the Cambridge Crystallographic Data Centre (CCDC), under deposition numbers CCDC 2050233, 2050080, and 2106067, respectively. These data can be obtained free of charge from The Cambridge Crystallographic Data Centre at www.ccdc.cam.ac.uk/data_request/cif.

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

## Acknowledgements

We gratefully acknowledge the support from the National Natural Science Foundation of China (22071079 for M.W.), the Natural Science Foundation of Jilin Province (20180101297JC for M.W.), and the staff from BL17B beamline of National Facility for Protein Science in Shanghai (NFPS) at Shanghai Synchrotron Radiation Facility for assistance during data collection.

## Author contributions

M.W. conceived and designed the experiments. J.M. and K.L. completed the synthesis. J.M., T.L., Y.X., Z.L., J.S., Q.B., Z.Z., X.-Q.H., Z.C. and P.W. conducted the characterization. J.M, T.L., X.D. and M.W. analysed the data and wrote the manuscript. All the authors discussed the results and commented on and proofread the manuscript.

## Competing interests

The authors declare no competing interests.
