## [Peer Review File · Communications Chemistry]

REVIEWERS' COMMENTS:

Reviewer #1 (Remarks to the Author):

The authors have made sound alterations too the manuscript and I'm happy to recommend that it is accepted for publication without further editing.

Reviewer #3 (Remarks to the Author):

The revised manuscript by Ma et al. has included some updated references and some more AFM micrographs in the Supporting Information, besides other minor changes. The AFM analysis is still not 100% clear to me (for example, the phase contrast in Figure S94 is very odd, as the inside of the cavity is very bright and the outside of the macrocycle basically presents the same contrast of the macrocycle). In this sense, I am somewhat disappointed that the authors did not take this opportunity to provide the original raw data for the referees to analyse.

All this said, I did not have any major issues with the original version and I am therefore happy to recommend acceptance of the manuscript in the present form.

Reviewer #4 (Remarks to the Author):

In the manuscript "Designing Narcissistic Self-Sorting Terpyridine Moieties with High Coordination Selectivity for Complex Metallo-Supramolecules" the authors report on the design of terpyridine based ligands with the purpose of selectively forming homoleptic complexes as a result of self-sorting. The figures and data reported are of a high quality. There is a wealth of characterisation data provided for all of the ligands and assembled species including ^1H , ^{13}C NMR and ESI-MS and these are well reported in the SI. The authors also report detailed ion mobility mass spectrometry data that, whilst not entirely necessary for most species, support the existence of the reported assemblies in solution.

I'm pleased to see that the vast majority of my previous concerns have been addressed and the findings are thoroughly examined and reported. This work, particularly the sections on selective self-sorting behaviour, will be of significant interest to the areas of supramolecular and coordination chemistry. I suggest publication of this article after the following points have been addressed:

1. The section on hierarchical self-assembly describes the formation of larger "nanostructures" observed by AFM and TEM. The exact structure of these assemblies is now presented as much more hypothetical, and rightly so given the evidence presented. However, there are some points where I think additional clarity would be helpful. The TEM images depicted in Fig 5b (and Figure S93) are too small to make it clear what the sizes of the complexes are relative to the scale bar. I suggest the authors add an enlarged inset and scale bar to make it clearer that the sizes are the 5.5 nm and 6.5 nm reported in the main text.
2. The term "cross-linked" is used to refer to the gels. Whilst I agree there is clearly aggregation, I would reserve the term "cross-linked" for specified covalent attachments.
3. The authors present evidence from the crystal structure that edge-to-edge stacking is solely responsible for the gelation behaviour. However, it seems just as likely that at higher concentrations,

the π - π stacking that drives self-sorting becomes less important and the system starts to assemble into coordination polymers. Do the authors have any evidence that coordination polymers do not occur? For instance, does the addition of terpyridine (tpy) effect the gelation concentration/temperature. If the structures are still self-sorting at these concentrations then tpy should have little effect on gelation. But if coordination polymers are responsible, tpy should break these up and prevent gelation. I do not think this information is necessarily required for publication but the potential for coordination polymers as a gelation mechanism should be discussed.

4. Concentrations should be included in Figure 6 or caption.

5. Where known, concentrations should also be included for NMRs given the dynamic nature of these systems.

6. The conclusions state “Moreover, these complexes showed significantly enhanced fluorescence intensity in solution state than the complexes based on conventional tpy”. Whilst this fact is true, the enhanced fluorescence is due largely to the additional ligand conjugation, as shown in figure S89. The formation of the complexes has little to do with this so this sentence is misleading.

Response letter to the reviewers' comments

Reviewers' comments:

Reviewer #1 (Remarks to the Author):

The authors have made sound alterations to the manuscript and I'm happy to recommend that it is accepted for publication without further editing.

Reply: We greatly appreciate the reviewer's positive comments on the work in this manuscript.

Reviewer #3 (Remarks to the Author):

The revised manuscript by Ma et al. has included some updated references and some more AFM micrographs in the Supporting Information, besides other minor changes. The AFM analysis is still not 100% clear to me (for example, the phase contrast in Figure S94 is very odd, as the inside of the cavity is very bright and the outside of the macrocycle basically presents the same contrast of the macrocycle). In this sense, I am somewhat disappointed that the authors did not take this opportunity to provide the original raw data for the referees to analyse.

Reply: We apologize for our misunderstanding of the previous reviewer's questions.

In response to the reviewer's comment, we provide the original raw data of Supplementary Figure 97 as follows:

Figure R1. The original raw data of AFM images of cyclic nanostructures formed by $\text{Zn}_9(\text{LA})_6$, (a) height, (b) amplitude error, (c) phase images of air-dried CH_3CN (10^{-5} M) dispersion on a silicon wafer substrate.

All this said, I did not have any major issues with the original version and I am therefore happy to recommend acceptance of the manuscript in the present form.

Reply: We thank the reviewer for the positive comments on our work.

Reviewer #4 (Remarks to the Author):

In the manuscript “Designing Narcissistic Self-Sorting Terpyridine Moieties with High Coordination Selectivity for Complex Metallo-Supramolecules” the authors report on the design of terpyridine based ligands with the purpose of selectively forming homoleptic complexes as a result of self-sorting. The figures and data reported are of a high quality. There is a wealth of characterisation data provided for all of the ligands and assembled species including ^1H , ^{13}C NMR and ESI-MS and these are well reported in the SI. The authors also report detailed ion mobility mass spectrometry data that, whilst not entirely necessary for most species, support the existence of the reported assemblies in solution.

I’m pleased to see that the vast majority of my previous concerns have been addressed and the findings are thoroughly examined and reported. This work, particularly the sections on selective self-sorting behaviour, will be of significant interest to the areas of supramolecular and coordination chemistry. I suggest publication of this article after the following points have been addressed.

Reply: We highly appreciate these encouraging remarks and thank the reviewer for highlighting important aspects of our work.

1) The section on hierarchical self-assembly describes the formation of larger “nanostructures” observed by AFM and TEM. The exact structure of these assemblies is now presented as much more hypothetical, and rightly so given the evidence presented. However, there are some points where I think additional clarity would be helpful. The TEM images depicted in Fig 5b (and Figure S93) are too small to make it clear what the sizes of the complexes are relative to the scale bar. I suggest the authors add an enlarged inset and scale bar to make it clearer that the sizes are the 5.5 nm and 6.5 nm reported in the main text.

Reply: We thank the valuable suggestion of the reviewer.

In response to the reviewer’s comment, we have modified the *Figure 6c*, *Figure 7b*, and *Figure 96A* in the resubmitted manuscript and Supporting Information.

Figure R2. Description of picture replacement in the resubmission.

Figure R3. Description of picture replacement in the resubmission.

Figure R4. Description of picture replacement in the resubmission.

2) The term “cross-linked” is used to refer to the gels. Whilst I agree there is clearly aggregation, I would reserve the term “cross-linked” for specified covalent attachments.

Reply: We thank the reviewer for this professional suggestion.

In response to the reviewer's comment, we have revised the manuscript by avoiding the term “cross-linked”, as “When the concentration further increased, these giant macrocycles formed a three-dimensional network via intermolecular interaction”.

3) The authors present evidence from the crystal structure that edge-to-edge stacking is solely responsible for the gelation behaviour. However, it seems just as likely that at higher concentrations, the π - π stacking that drives self-sorting becomes less important and the system starts to assemble into coordination polymers. Do the

authors have any evidence that coordination polymers do not occur? For instance, does the addition of terpyridine (tpy) effect the gelation concentration/temperature. If the structures are still self-sorting at these concentrations then tpy should have little effect on gelation. But if coordination polymers are responsible, tpy should break these up and prevent gelation. I do not think this information is necessarily required for publication but the potential for coordination polymers as a gelation mechanism should be discussed.

Reply: We thank the valuable suggestion of the reviewer.

We conducted the experiment according to the reviewer's suggestion, and the addition of **Zn(tpy)₂** destroyed the gel formation at a very high concentration (~ 40 mg/mL). However, when we mixed **Zn₉(LA)₆** and **Zn(tpy)₂** together at a low concentration (4.5 mg/mL), some new NMR signals appeared which indicated that the hexagon was also destroyed by **Zn(tpy)₂** (Figure R5). Besides, we also tried other methods like measuring diffusion-ordered NMR spectroscopy (DOSY) of **Zn₉(LA)₆** at high concentrations, however, the result was still less convincing. Actually, it is difficult to prove whether the gel was formed by coordination polymers or macrocycles at very high concentrations, or both of them existed, but we tend to think the macrocycles play a major role according to the reported literatures (please see *J. Am. Chem. Soc.* **2018**, *140*, 3257-3263; *J. Am. Chem. Soc.* **2015**, *137*, 1556-1564.).

Figure R5. ¹H NMR spectra (500 MHz, CD₃CN, 300 K) of Zn(tpy)₂ (top), Zn₉(LA)₆ (bottom) and the mixture of both (4.5 mg/mL).

4) Concentrations should be included in Figure 6 or caption.

Reply: We thank the reviewer for the instructive comment.

In response to the reviewer's comment, we have illustrated the concentrations in caption of *Figure 6* and *Figure 7* in the resubmitted manuscript.

“Fig. 6 Characterization of hierarchical self-assembly of Zn₉(LA)₆. **a** Schematic diagram of edge-to-edge stacking by Zn₉(LA)₆. **b** The energy-minimized structure of Zn₉(LA)₆ from molecular modeling. **c** TEM images of individual Zn₉(LA)₆ at a concentration of 10⁻⁶ M in CH₃CN (scale bar, 50 nm and 20 nm for zoom-in image). **d**

TEM images of the assembled cyclic nanostructures of $\text{Zn}_9(\text{LA})_6$ at a concentration of 10^{-5} M in CH_3CN (scale bar, 50 nm and 20 nm for zoom-in image). **e** TEM images of the 3D network structure formed by assembled cyclic nanostructures of $\text{Zn}_9(\text{LA})_6$ at a concentration of 10^{-4} M in CH_3CN (scale bar, 100 nm). **f** AFM images of the hierarchically assembled cyclic nanostructures by $\text{Zn}_9(\text{LA})_6$ (scale bar, 50 nm). **g** Cross-section of the cyclic nanostructure shown in image **f**.”

“**Fig. 7 Characterization of the hierarchical self-assembly of $\text{Zn}_{12}(\text{LB})_6$.** **a** The energy-minimized structure of $\text{Zn}_{12}(\text{LB})_6$ from molecular modeling. **b** TEM images of individual $\text{Zn}_{12}(\text{LB})_6$ at a concentration of 10^{-6} M in CH_3CN (scale bar, 50 nm and 20 nm for zoom-in image). **c** TEM images of the assembled cyclic nanostructures of $\text{Zn}_{12}(\text{LB})_6$ at a concentration of 10^{-5} M in CH_3CN (scale bar, 100 nm). **d** TEM images of the 3D network structure formed by assembled cyclic nanostructures of $\text{Zn}_{12}(\text{LB})_6$ at a concentration of 10^{-4} M in CH_3CN (scale bar, 100 nm). **e** Photograph of $\text{Zn}_{12}(\text{LB})_6$ gel formation at a concentration of 25 mg/mL in CH_3CN .”

5) Where known, concentrations should also be included for NMRs given the dynamic nature of these systems.

Reply: We thank the reviewer for this suggestion.

In response to the reviewer’s comment, we have illustrated the concentrations for NMRs of these assemblies in *Figure 2b*, *Figure 3b*, *Figure 5a*, *Figure 5d*, *Supplementary Figs. 19-22, 89, and 90*.

6) The conclusions state “Moreover, these complexes showed significantly enhanced fluorescence intensity in solution state the complexes based on conventional tpy”. Whilst this fact is true, the enhanced fluorescence is due largely to the additional ligand conjugation, as shown in figure S89. The formation of the complexes has little to do with this so this sentence is misleading.

Reply: We thank the reviewer for the insightful comments.

We fully agree with the reviewer that the expression in our previous manuscript could be misleading. According to the results of the fluorescence data, the emission intensity of ligands **MA**, **MB**, **LA**, and **LB** increased to the 5.0-fold, 5.5-fold, 4.4-fold, and 5.8-fold compared with that of **tpy**, respectively, due to the additional ligand conjugation. Notably, the emission intensity of complexes $\text{Zn}_2(\text{MA})_2$, $\text{Zn}_3(\text{MB})_2$, $\text{Zn}_9(\text{LA})_6$, and $\text{Zn}_{12}(\text{LB})_6$ increased to the 6.2-fold, 8.6-fold, 8.8-fold, and 11.0-fold compared with that of $\text{Zn}(\text{tpy})_2$, respectively, and this additional enhanced fluorescence are attributed to the restricted chemical environment of supramolecular structures.

In response to the reviewer's comment, we have modified the sentence as: “Moreover, these complexes showed significantly enhanced fluorescence intensity in solution state than the complexes based on conventional tpy due to the additional ligand conjugation and the restricted chemical environment.”